

# The effectiveness of Payments for Ecosystem Services at delivering improvements in water quality: lessons for experiments at the landscape scale

Edwin L. Pynegar[1,2], Julia P.G. Jones[1], James M. Gibbons[1] and Nigel M. Asquith[3,4]

[1] School of Environment, Natural Resources and Geography, Bangor University, Bangor, United Kingdom
[2] School of Life Sciences, University of Sussex, Falmer, United Kingdom
[3] Harvard Forest, Harvard University, Petersham, United States of America
[4] Kennedy School of Government, Harvard University, Cambridge, United States of America

Corresponding author
Julia P.G. Jones,
julia.jones@bangor.ac.uk

## ABSTRACT

**Background**. Randomised Control Trials (RCTs) are used in impact evaluation in a range of fields. However, despite calls for their greater use in environmental management, their use to evaluate landscape scale interventions remains rare. Payments for Ecosystem Services (PES) incentivise land users to manage land to provide environmental benefits. We present the first RCT evaluation of a PES program aiming to improve water quality. *Watershared* is a program which incentivises landowners to avoid deforestation and exclude cattle from riparian forests. Using this unusual landscape-scale experiment we explore the efficacy of *Watershared* at improving water quality, and draw lessons for future RCT evaluations of landscape-scale environmental management interventions.

**Methods**. One hundred and twenty-nine communities in the Bolivian Andes were randomly allocated to treatment (offered *Watershared* agreements) or control (not offered agreements) following baseline data collection (including *Escherichia coli* contamination in most communities) in 2010. We collected end-line data in 2015. Using our end-line data, we explored the extent to which variables associated with the intervention (e.g. cattle exclusion, absence of faeces) predict water quality locally. We then investigated the efficacy of the intervention at improving water quality at the landscape scale using the RCT. This analysis was done in two ways; for the subset of communities for which we have both baseline and end-line data from identical locations we used difference-in-differences (matching on baseline water quality), for all sites we compared control and treatment at end-line controlling for selected predictors of water quality.

**Results**. The presence of cattle faeces in water adversely affected water quality suggesting excluding cattle has a positive impact on water quality locally. However, both the matched difference-in-differences analysis and the comparison between treatment and control communities at end-line suggested *Watershared* was not effective at reducing *E. coli* contamination at the landscape scale. Uptake of *Watershared* agreements was very low and the most important land from a water quality perspective (land around water intakes) was seldom enrolled.

**Discussion**. Although excluding cattle may have a positive local impact on water quality, higher uptake and better targeting would be required to achieve a significant impact on the quality of water consumed in the communities. Although RCTs potentially have an important role to play in building the evidence base for approaches such as PES, they are far from straightforward to implement. In this case, the randomised trial was not central to concluding that *Watershared* had not produced a landscape scale impact. We suggest that this RCT provides valuable lessons for future use of randomised experiments to evaluate landscape-scale environmental management interventions.

# INTRODUCTION

Whether an intervention is effective at delivering the outcomes expected is a key question for evidence-based policy making (e.g., *White, 2013*). This question is highly pertinent in the field of conservation and environmental management as awareness grows of the amount of money that has been spent on interventions with limited understanding of their effectiveness (*Ferraro & Pattanayak, 2006*; *Bowler et al., 2012*; *Waeber et al., 2016*; *Salzman et al., 2018*). As a result, conservation effectiveness is being widely discussed in the academic literature (*Baylis et al., 2016*; *Börner et al., 2016*; *Sutherland & Wordley, 2017*), the policy literature (*Puri et al., 2016*; *Duchelle, Wunder & Martius, 2018*), conservation journalism (*Dasgupta, Gaworecki & Burivalova, 2018*) and the mainstream media (*Mooney, 2016*). Randomised Control Trials (RCTs), in which experimental units are randomly allocated to treatment or control groups, allow the creation of robust counterfactuals from which to infer what would have happened in the absence of the intervention (e.g. *Rubin, 1974*). RCTs are widely used in many areas of public policy including medicine, education, and development economics (*Glennerster & Takavarasha, 2013*; *Council of Economic Advisers, 2014*). Although small-scale RCTs have been a mainstay of applied ecological experiments for decades, there are very few examples of RCTs of large-scale environmental management interventions, and there have been calls for their increased use (*Greenstone & Gayer, 2009*; *Samii et al., 2014*; *Baylis et al., 2016*; *Börner et al., 2016*; *Börner et al., 2017*).

Payments for Ecosystem Services (or Payments for Environmental Services—the terms are largely interchangeable (*Wunder, 2015*)) translate external, non-market values of the environment into financial incentives for local actors to provide environmental services. The focus of many PES programs in Latin America (*Martin-Ortega, Ojea & Roux, 2013*; *Grima et al., 2016*), and to a lesser extent in Asia and Africa, e.g., (*Calvet-Mir et al., 2015*) is the increase or maintenance of supply of good quality water. At least 1.8 billion people still rely on drinking water sources contaminated with faecal matter (*Bain et al., 2014a*). Where sources lack adequate physical or chemical treatment the quality of drinking water is influenced by land use and ecosystem management around and upstream of those water

sources. Hence provision of clean water can be considered as an ecosystem service or as a precursor to multiple ecosystem services benefiting society (*Keeler et al., 2012*). Despite the increasing number of PES programs in operation, there are very few robust evaluations of the extent to which they deliver the outcomes they seek to supply (*Pattanayak, Wunder & Ferraro, 2010*; *Miteva, Pattanayak & Ferraro, 2012*; *Naeem et al., 2015*; *Baylis et al., 2016*; *Börner et al., 2017*; *Ferraro, 2017*; *Salzman et al., 2018*). We know of only a single published Randomised Control Trial of a PES (evaluating the impact of a program in Uganda on deforestation; *Jayachandran et al., 2017*) and none evaluating the impact of PES on water quality.

Gastrointestinal illnesses caused by consumption of contaminated water are a major cause of mortality and morbidity in the developing world (*Prüss-Ustün et al., 2014*). *Escherichia coli* is a bacterium that lives only in the guts of warm-blooded animals (*Leclerc et al., 2001*). While some strains of *E. coli* are pathogenic, the majority are not but are useful indicators of faecal contamination and the presence of other pathogens (*Ashbolt, Grabow & Snozzi, 2001*). Sources of faecal contamination may include faulty sewerage systems and leaking septic tanks (*Richards et al., 2016*), open defecation (*Spears, Ghosh & Cumming, 2013*), or the presence of wildlife (*Ahmed et al., 2012*). However, a major source of contamination is the presence of domestic livestock, particularly free-roaming cattle (*Crane et al., 1983*). Therefore, cattle exclusion has been practiced as a means of reducing faecal contamination of watercourses. In the UK, for example, the Good Agricultural and Environmental Conditions standard 1 requires farmers in receipt of certain subsidies to maintain buffer strips and refrain from spreading manure within areas close to water bodies (*GOV.UK, 2016*). There is evidence of such actions being effective at significantly reducing *E. coli* concentration and other faecal contamination of water supplies (*Sunohara et al., 2012*). However, many uncertainties remain about the extent to which these interventions, incentivised via a PES program, can deliver consistent benefits in water quality at the landscape scale.

The Bolivian non-governmental organization *Fundación Natura Bolivia* (*Natura*) began using in-kind incentives to encourage conservation in the Andean region of Bolivia in 2003. Their program, now known as *Watershared,* aims to slow forest loss and protect the quality of water available to communities through providing modest development support in exchange for avoiding deforestation and excluding livestock from riparian forest (*Bottazzi et al., 2018*). Although *Natura* does not characterise *Watershared* as PES (*Asquith, 2016*), the program meets the most widely used PES definition (*Wunder, 2015*): "voluntary transactions between service users and service providers that are conditional on agreed rules of natural resource management for generating offsite services". As of 2016, 210,000 hectares of forest owned by 4,500 households were under *Watershared* conservation agreements (*Asquith, 2016*).

Given the growing interest in evaluating the effectiveness of different conservation approaches, *Natura* established a Randomised Control Trial (RCT) to evaluate *Watershared*. One hundred and twenty-nine communities were randomly allocated to control (not offered *Watershared* agreements) or treatment groups (offered agreements). We use this unique setup to investigate the effectiveness of the intervention at delivering improvements

in microbial water quality. We address three interconnected questions: (1) Do the features of *Watershared* agreements (e.g., cattle exclusion, absence of faeces) have a measurable impact on water quality at a site, accounting for other predictors? (2) Did the implementation of *Watershared* in treatment communities result in an improvement in water quality relative to control communities? (3) What lessons does the *Watershared* RCT evaluation offer for the wider use of experiments to evaluate the impact of conservation interventions at the landscape scale?

## MATERIALS AND METHODS

### Context and RCT design

This article focuses on the *Watershared* intervention in the Río Grande Valles Cruceños Natural Integrated Management Area (Spanish acronym *ANMI Río Grande-Valles Cruceños*), a protected area of 7,339 km$^2$ in the Andean region of the Santa Cruz Department in eastern Bolivia (Fig. 1). Forests in this area are perceived locally as contributing to providing high-quality water for human consumption and irrigation, despite the mixed scientific evidence on this topic (*Bruijnzeel, 2004*; *Ponette-González et al., 2015*). Gastrointestinal illnesses are endemic; for example, in 2015 the health centre of Moro Moro, a community of approximately 800 people, treated 236 cases of diarrhoea (information from *Servicio Nacional Integral de Salud, Centro de Salud Moro Moro*, obtained 4th April 2016). Faecal contamination from cattle is widely considered an important contributor to the high prevalence of these diseases as the traditional farming system involves cattle grazing freely within the forests from where most communities take their water. While some communities have rudimentary sedimentation and filtration systems, these are of limited effectiveness and often become clogged with sediment after each rainfall event. Chlorination or other chemical treatment is rare.

In 2010, 129 communities within the *Río Grande-Valles Cruceños* protected area were selected for inclusion in a Randomised Control Trial (Fig. 1). Consent to randomisation was granted by community leaders on the understanding that the intervention would subsequently be implemented in all communities (this general roll-out was conducted in 2016). Communities were randomly allocated to control (64 of these communities in which conservation agreements were not offered) or treatment (65 communities in which agreements were offered) groups following stratification based on municipality, community size, and estimated cattle density. The RCT was not blinded as participants unavoidably knew whether they belonged to a treatment or control community. However, in order to avoid observer bias effects during data collection, those conducting water quality monitoring did not know which communities belonged to the treatment or control group.

Individuals belonging to treatment communities were offered the chance to conserve land belonging to them under *Watershared* agreements (see *Bottazzi et al. 2018* for more detail on the *Watershared* program) and received education on the importance of cattle exclusion and forest conservation for the maintenance of water quality and quantity. Individuals belonging to control communities received the environmental education only. *Natura* offered landowners in treatment communities three-year conservation agreements
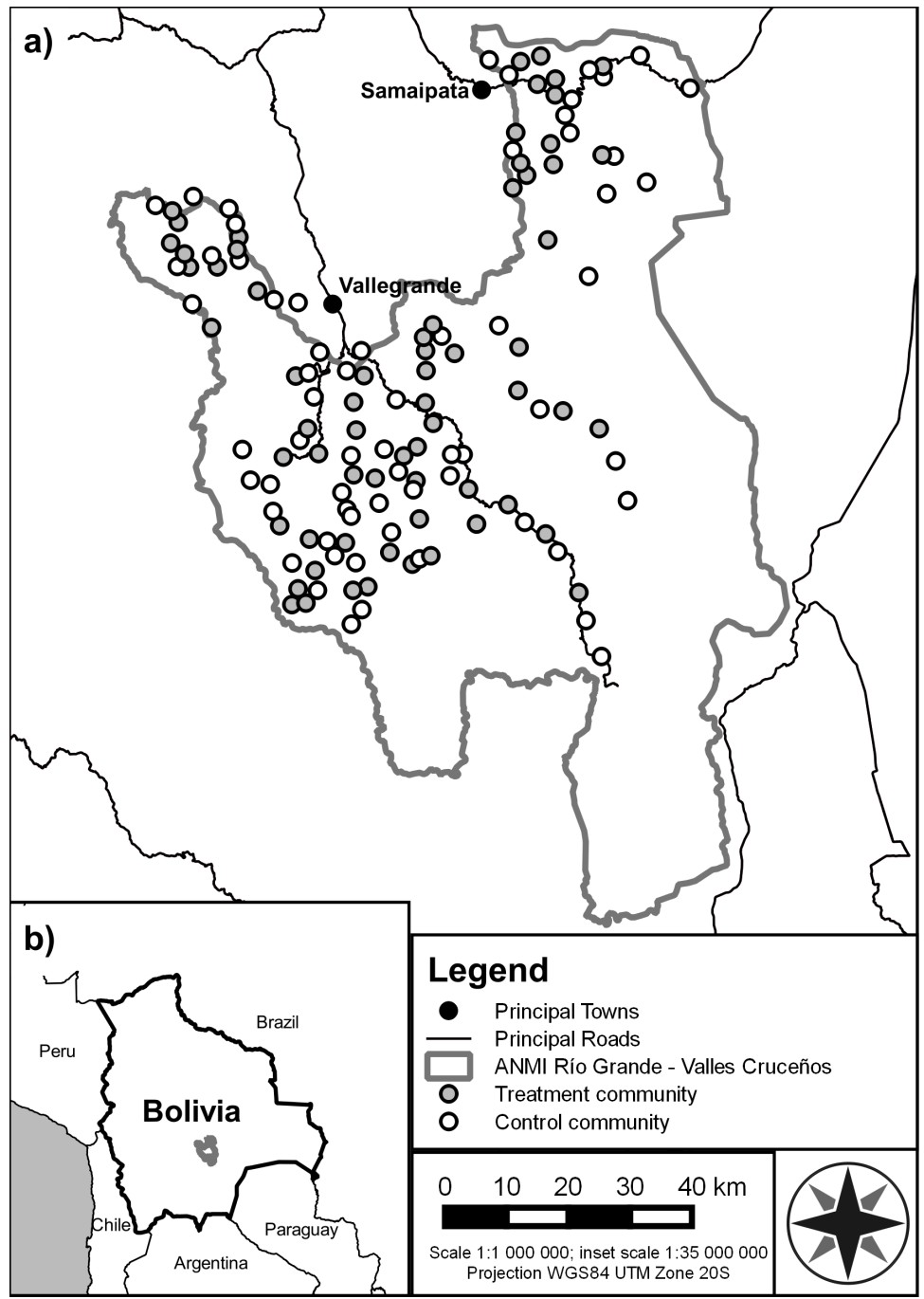

**Figure 1  Map of the study area.** (A) Locations of the 65 treatment communities (*Watershared* agreements offered) and 64 control communities (*Watershared* agreements not offered) within the *Río Grande Valles Cruceños* Natural Integrated Management Area (*ANMI RG-VC*). (B) Location of the ANMI RG-VC protected area within Bolivia.

to conserve upstream forest and exclude cattle from riparian forest in return for in-kind incentives such as fruit trees, barbed wire, or irrigation materials. Participants could enrol their land in one of three kinds of agreements (for details see Table S1). In this paper we only consider level 1 agreements, in which landowners were offered $10/hectare/year in-kind equivalent (plus the equivalent of $100 regardless of the size of the area enrolled) in return for conserving forested land within 100 m of a watercourse and excluding cattle from these areas. Landowners were offered the opportunity to enrol their land twice per year, beginning in August 2011. Compliance monitoring and distribution of the in-kind compensations was conducted yearly. A recent analysis suggests that 31% of the area of level 1 agreements resulted in additional conservation (i.e., cattle were kept out of land which otherwise they would have been allowed in; Bottazzi et al., 2018).

## Sampling strategy

Our analyses are based upon two rounds of monitoring of the quality of water intended for human consumption. A baseline was taken between February and July of 2010 by the NGO *Natura* before the sites were allocated to control or treatment groups. The allocation to control or treatment was not stratified by measured water quality at the sites, and the baseline data was not otherwise used until our team started work on the project in 2013. A more detailed end-line monitoring round was undertaken by our team from Bangor University in collaboration with Natura between March and May of 2015, i.e., following completion of the first signed agreements. In the end-line we had more stringent protocols and also measured a number of additional potential indicators of water quality.

The communities within the RCT are small (maximum number of households is 123; Bottazzi et al., 2017) with diverse water supply systems. Some have a single water intake, others multiple intakes and in a few cases no functional intake at all (community members take water directly from streams or other water bodies). Resource and logistical constraints meant that not all intakes and taps could be sampled and so the tap supplying the community's school, along with the intake supplying that tap, were taken as sampling sites based on the assumption that these would have the greatest importance for health (Fig. 2). In cases where the community had no school, we monitored at the intake which supplied the greatest number of households and a representative tap fed by that intake. In the cases in which the community had no functional water system at all, we took a sample in the water body where the greatest number of households collected their water. Thus most communities had two site measurements (intake and tap), whereas a few (those lacking an intake) only had one, in a few communities two intakes were measured. In this paper we refer to the combination of an intake and a tap (or the location where households collected water where an intake is not present) as a water system.

In 2010 the randomisation process assigned 129 communities to control or treatment group. Independently technicians from *Natura* monitored water quality in 120 of these communities. In the 2015 end-line we monitored 118 communities that were part of the RCT plus an additional six communities not in the RCT. Two of the RCT sites were excluded from the end-line analysis as their water system was supplied from rainfall collected from roofs and therefore cattle could not affect water quality. Water quality was

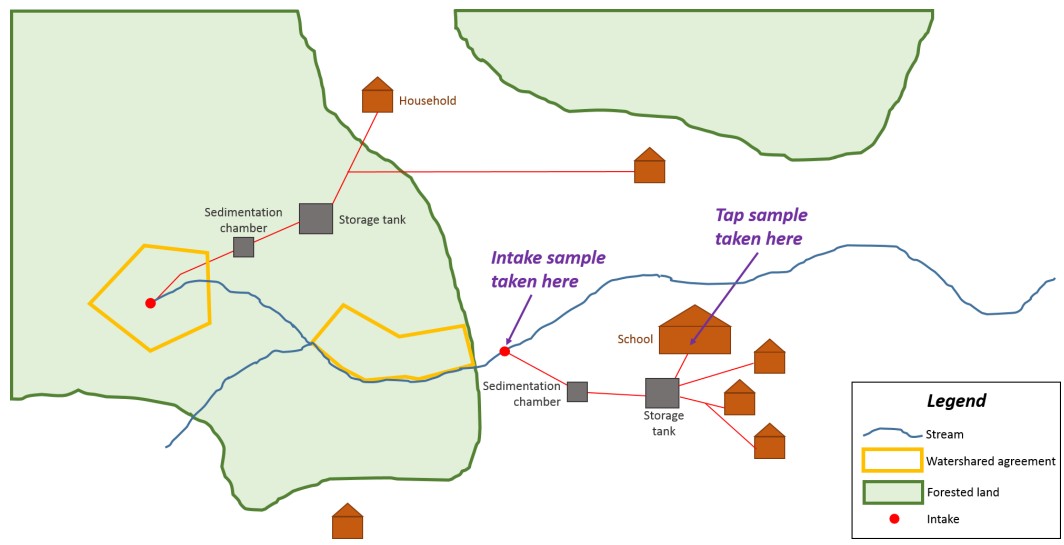

**Figure 2** Schematic of an example community with two water intakes showing locations of intake and tap sampling sites.

measured at each site (see below). In 2015 we also recorded other characteristics of the site which may predict water quality. Table S2 summarises the sample size for each analysis.

The *Natura* technicians who took water measurements in 2010 recorded the location with a handheld GPS. However when we returned in 2015 it was not always possible to confirm that we were at the same water intake (GPS readings in forest can be inaccurate). In addition, between baseline data collection in 2010 and our team's visit in 2015, the location of both water intakes and the main tap serving the community had changed in a number of sites. As a result, for only 83 sites in 47 communities were we 100% confident that had measured water quality at the same location between 2010 and 2015 (Table S2). The fact that end-line data from the same location was not available from all sites is somewhat analogous to the common problem of attrition due to loss to follow up in clinical trials (*Jüni, Altman & Egger, 2001*). However as we still took an end-line measure (although as we could not be sure the location was identical and therefore the data were not included in the difference–in-differences analysis) we do not use the term attrition in this paper. It is also important to note that a few of the water intakes could not be unambiguously assigned to control or treatment (for example there are two cases in which one intake supplied two communities, one of which was treatment and the other control); these have been excluded from the RCT analysis.

## Water quality monitoring

The principal metric recorded was *E. coli* colony forming unit concentration (CFUs) in water samples. *E. coli* concentration, along with that of other non-*E. coli* bacteria belonging to the coliform group, was quantified using the Coliscan Easygel method (Micrology Labs, Goshen, IN, USA). Coliscan Easygel allows enumeration of coliforms as after incubation *E. coli* colonies appear purple, blue-purple or dark blue due to metabolism of both

beta-galactosidase and beta-glucuronidase. Other non-*E. coli* coliforms are pink based upon metabolism of beta-galactosidase only. Colonies of a blue–green or sky blue color (metabolism of beta-glucuronidase only) and white colonies were not counted (*Micrology Labs, 2016*). The Easygel method (which uses only five ml of water per sample) does not comply with the World Health Organization's 100 ml standard for coliform monitoring but studies have shown that it is reasonably robust and not susceptible to false negatives (*Chuang, Trottier & Murcott, 2011*). This method had been selected by *Natura* due to the logistical challenges with using alternative methods such as membrane filtration in the remote and low-resource context of the study area (many sites are reachable only with difficult drives and long walks). When we modified the protocol in 2015 (to overcome some of the limitations of the 2010 protocol), we elected to retain the method for the same reason and to ensure data were comparable as possible.

In 2010, one sample was placed into sterile Coliscan Easygel sampling flasks (35 ml) taking care to avoid any external contamination. Up to two days later (but normally on the same day) the *Natura* team then inoculated Easygel Petri dishes using 5 ml of the water from each flask. After solidification the Petri dishes were sealed and incubated at ambient temperature for 48 h, after which numbers of *E. coli* and other non-*E. coli* coliform CFUs were counted.

In 2015, four separate samples were taken using sterile Coliscan Easygel sampling flasks (35 ml each) and placed on ice within 1 h of sampling. Within 6 h of sampling (although generally within 4) we produced Easygel Petri dishes using five ml of water from each flask as inoculum. After solidification we sealed the Petri dishes and incubated them for 24 h at 35–37 °C in a portable incubator (NQ28 model, Darwin Chambers, St Louis, MO, USA). In locations where no mains electricity was available we maintained a constant incubation temperature through use of a 12 V vehicle power supply or supply from a car battery. After incubation we counted *E. coli* and other non-*E. coli* coliform CFUs.

In 2015 we also measured in each site a number of physico-chemical parameters of water: temperature, dissolved oxygen in mg/l and per cent of saturation value, pH, salinity and conductivity in each site with an HQ40d portable multi-parameter meter and IntelliCAL LDO101, PHC101 and CDC401 rugged probes respectively (HACH Company, Loveland, CO, USA). We measured turbidity in formazin attenuation units through the use of a DR/850 colorimeter (HACH Company, Loveland, CO, USA). Additionally, at the intake sites, we recorded other variables that may predict *E. coli* concentration, including the presence or absence of cattle (judged based upon presence of faeces, hoof prints, or cattle paths recently used) and the presence or absence of cattle faeces in the riparian forest, in the water, or on banks. Some were recorded at the intake itself and others along a 10 m transect upstream (uphill in the case of intakes in springs) of the intake. Details of all monitored variables are available in Table S3.

We used *Natura's* community database to determine which intakes supplied treatment or control communities (we did not have this information when conducting field sampling to avoid any observer bias effects). We used GIS software (ArcGIS 10.2, ESRI, Redlands, CA, USA) and *Natura's* shapefiles to calculate the percentage of eligible land in each community which was enrolled in *Watershared* agreements (we compared this percentage

between treatment and control communities), and which monitored intakes fell within land enrolled in *Watershared* agreements. We used data held by *Natura* to confirm which of these intakes were in agreements which had been compliant with the agreements according to Natura's observations. In a number of the earliest sites monitored during 2015 we accidentally disturbed the sediment in the water intake while taking samples; sites in which this happened were recorded as such.

## Balance achieved in the allocation of control and treatment

At baseline *Natura* collected a number of variables at the community level. Two of these (mean number of cows per person in the community, and number of households in the community; both blocked into two groups) were used in the stratification to allocate communities to control or treatment. We explored the balance achieved at baseline for measured variables which may affect the outcome of interest (*E. coli* contamination at end-line) using standardized mean differences between treatment and control estimated in the R cobalt package (*Greifer, 2018*). Included variables were those used in the stratification, time from community centre to hospital (a proxy for remoteness), turbidity and the baseline measure of *E. coli* water contamination. Mean differences were standardized using pooled standard deviations (Fig. 3). Looking at all 120 communities monitored at baseline, variables appear quite well balanced (all close to or <0.25 pooled standard deviation units).

However, there is less balance between control and treatment at baseline when just the sites for which we have comparable data for baseline and end-line (i.e., excluding all sites for which we were not 100% sure that water quality was measured at the same location). When the baseline conditions for these 83 sites in 47 communities are compared, it is clear an imbalance has been introduced in a number of variables. In particular, the remaining control sites had substantially higher *E. coli* contamination) at baseline and were in more remote communities.

## Statistical analysis

We used generalized linear mixed model (GLMMs) to investigate whether features of *Watershared* agreements (e.g., cattle exclusion, absence of faeces) have a measurable impact on water quality at a site accounting for other predictors. For this we used the much richer 2015 data. We used the glmmADMB package in R (*Fournier et al., 2012*; *R Development Core Team, 2014*) to produce GLMMs predicting *E. coli* concentrations, specifying a negative binomial error structure and log-link. We included the water system identifier throughout as a random effect, as measurement at an intake and then a tap supplied by that intake represents repeated measures of the same water system. The unit of analysis is therefore the water system. We used model selection based upon comparisons of the Akaike's Information Criterion (AIC) and compared relative goodness of models through Akaike weighting. We then determined 95% confidence intervals for predictors in the principal model of interest. For this analysis we used all sites monitored which had a complete set of predictors, with the exception of a single site where the community collect water from a river with a catchment size of 9,768 km$^2$, meaning this site is qualitatively different from all other sites (water systems $N = 124$). For predictor variable selection

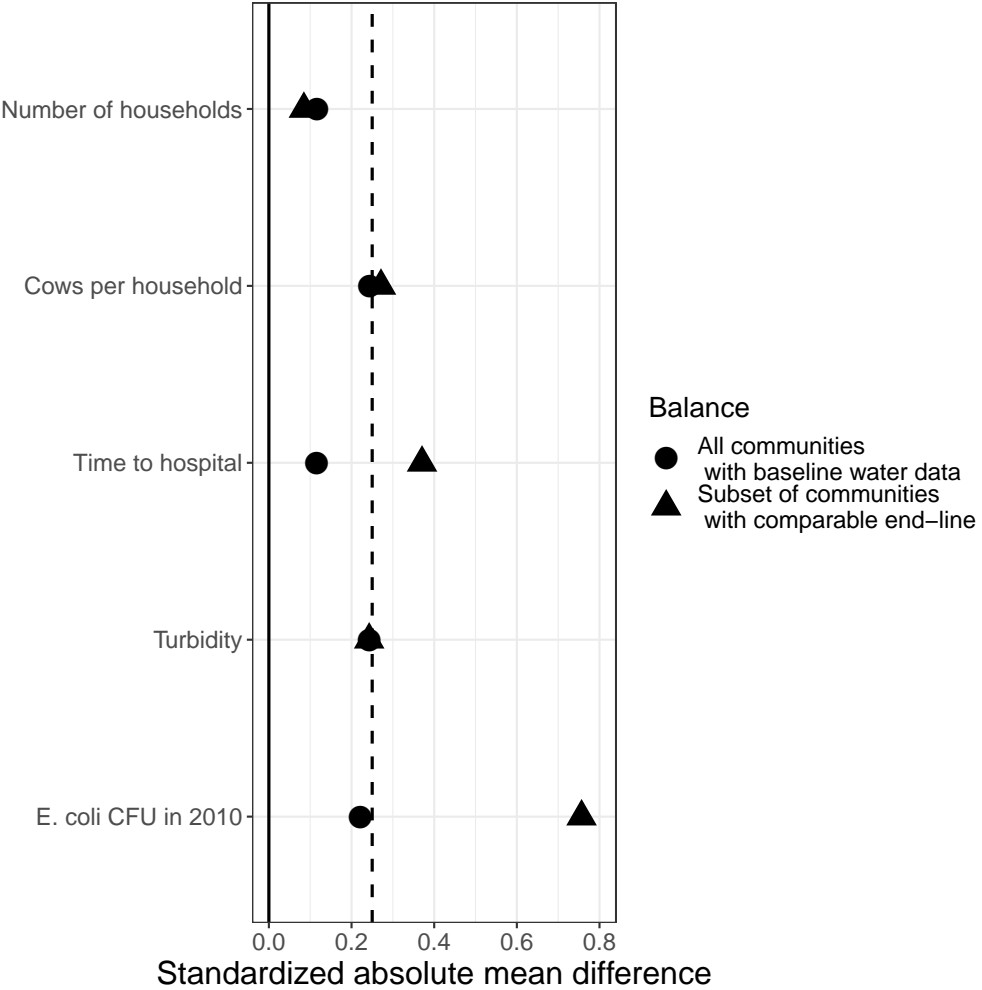

**Figure 3 Balance plot showing the standardized absolute mean differences between control and treatment communities at baseline.** Key baseline variables likely to influence end-line microbial water quality are shown both for the 120 communities where water quality was monitored at baseline, and the subset of 47 communities for which the monitoring location remained the same between baseline and end-line data collection. The loss of sites resulted in a substantial increase in imbalance for two variables (especially in baseline *E. coli* contamination). The dashed line indicates acceptable balance (<0.25 pooled standard deviation units).

we first removed closely correlated predictors and then classified variables we considered likely to be important in predicting *E. coli* concentration (Table 1). We associated predictor data relating to intake features with both intakes and their respective associated taps. We then produced GLMMs for all purely biophysical traits of sites, while also including all two-way and three-way interactions between temperature, pH, and salinity (Table S4A). To determine intervention effectiveness, we then added features that related directly to the intervention (cattle access, whether the site was in a level 1 *Watershared* agreement, and faeces presence) and again conducted model selection based on AIC minimization (See Table S4B).
**Table 1 Variables hypothesized to be important in predicting _E. coli_ concentration in 2015.** Codes are used in subsequent model selection tables (Tables S4A and S4B).

| Variable | Code | Classification | Base level |
|---|---|---|---|
| Site type | ST | Intake; Tap | Intake |
| Intake category | IC | Stream; Spring | Stream |
| Sediment disturbance | SD | Undisturbed; Disturbed | Undisturbed |
| Intake substrate | IS | Rock only; with sand; with mud | Rock only |
| Cattle presence | C | Absent; Present | Absent |
| Agriculture presence | A | Absent; Present | Absent |
| Turbidity | Tu | Continuous; FAU/100 | – |
| Temperature | T | Continuous | – |
| Salinity | S | Continuous | – |
| pH | pH | Continuous | – |
| Cattle access | CA | Yes; No | Yes |
| Faces presence | F | Absent; Present in forest; Present in water or on stream banks | Absent |
| Compliant level 1 _Watershared_ area | ARA | None; Intake entirely within conserved area | None |

We used two approaches to explore whether the implementation of _Watershared_ in treatment communities resulted in an improvement in water quality relative to control communities. In the first approach (evaluating the difference-in-differences from 2010 and 2015 between treatment and control sites), we only included the subset of sites where the sample locations remained the same between 2010 and 2015 and the intake is unambiguously associated with a treatment or control community (site $N = 83$, communities $N = 47$). While we use the term difference-in-differences analysis to refer to this analysis, we note that we do not use the standard difference–in-differences estimator, but take a panel approach and include the baseline outcome value as a covariate (_Gelman & Hill, 2007_). This approach is considered more appropriate especially if autocorrelation is low and has greater power (_McKenzie, 2012_). As the balance check suggested that the reduction in sites had resulted in a lack of balance in pre-existing contamination we performed a matched analysis. Sites were matched on the baseline _E. coli_ measure using genetic matching in the R MatchIt package (_Ho et al., 2011_). Using the weighting from this matched data we then estimated a differences-in-differences GLMM with _E. coli_ concentration in 2015 as the response variable and site treatment status (whether a site is in a control or treatment community) and 2010 _E. coli_ concentration as potential predictors. Retention of the 2010 _E. coli_ count controlled for the different pre-existing levels of contamination within the matched data set. We also included an interaction term between 2010 _E. coli_ concentration and site treatment status (if this interaction were a significant predictor, this would represent a significant effect of the intervention on water quality). Given the different volume of water sampled in 2010 (five ml) and 2015 (20 ml), we included an offset term of $\log_e(4)$ in each of the models to ensure equivalence between

2010 and 2015 *E. coli* CFU counts. We also included water system as a random effect to account for the tap and intake measures. This model was fitted using the glmmTMB package in R (*Brooks et al., 2017*) as glmmADMB does not accommodate weights. To account for taps and intakes being connected we included water system as a random effect.

In the second approach (evaluating at end-line only) *E. coli* concentrations in 2015 for all sites (site $N = 228$, community $N = 116$) measured were used (regardless of whether we had baseline water quality values measured at the identical location). Again the model was fitted in glmmTMB with site treatment status as the predictor of interest but also including the water source (spring or stream) and data collection point (tap or intake) to control for any differences in these. For both analyses we excluded two sites where water was collected from roofs rather than streams or springs. We accounted for some communities having more than one water system measured by including water system and community as a random effect.

To establish whether in practice land use differs between treatment and control communities, we also determined (for all intakes monitored in 2015 for which data on cattle access is available) whether relative proportions of intake sites protected from cattle differed between the treatment and control communities. We tested for a significant difference using a chi-squared test.

## RESULTS

### Cattle faeces in water is one of the significant predictors of *E. coli* concentration at the local scale

*E. coli* concentration in 2015 (water systems $N = 124$) is significantly predicted by a number of variables (Fig. 4). The details of model selection can be seen in Table S4A (for purely biophysical model selection) and Table S4B for model selection including parameters relating directly to the intervention. Intakes are significantly more contaminated than taps, sites associated with stream intakes are significantly more contaminated than sites associated with spring intakes, and turbidity and disturbance of the sediment by the research team during sampling are both also associated with higher recorded contamination. In terms of variables directly connected to the intervention, the presence of cattle faeces in or close to the water is a significant predictor of contamination. Although faeces presence in the wider forest shows a positive trend, it is not significant at 95% CI. Details of the model can be found in Table S4C.

### The intervention had no significant effect on *E. coli* concentration at the landscape scale

We analyze the RCT in two ways. In the first (difference-in-differences; Fig. 5) we used weights derived from genetic matching the communities based on the *E. coli* count in 2010 and only the sub-sample of communities where we are certain water sampling locations are the same in 2010 and 2015 (sites $N = 83$, communities $N = 47$). The weighting strongly downweighted two communities ($w = 0.19$), moderately upweighted nine ($w = 1.19$), strongly upweighted one ($w = 1.5$) and did not change the relative weighting of the remaining communities. We include *E. coli* count in 2010 as a predictor to control for

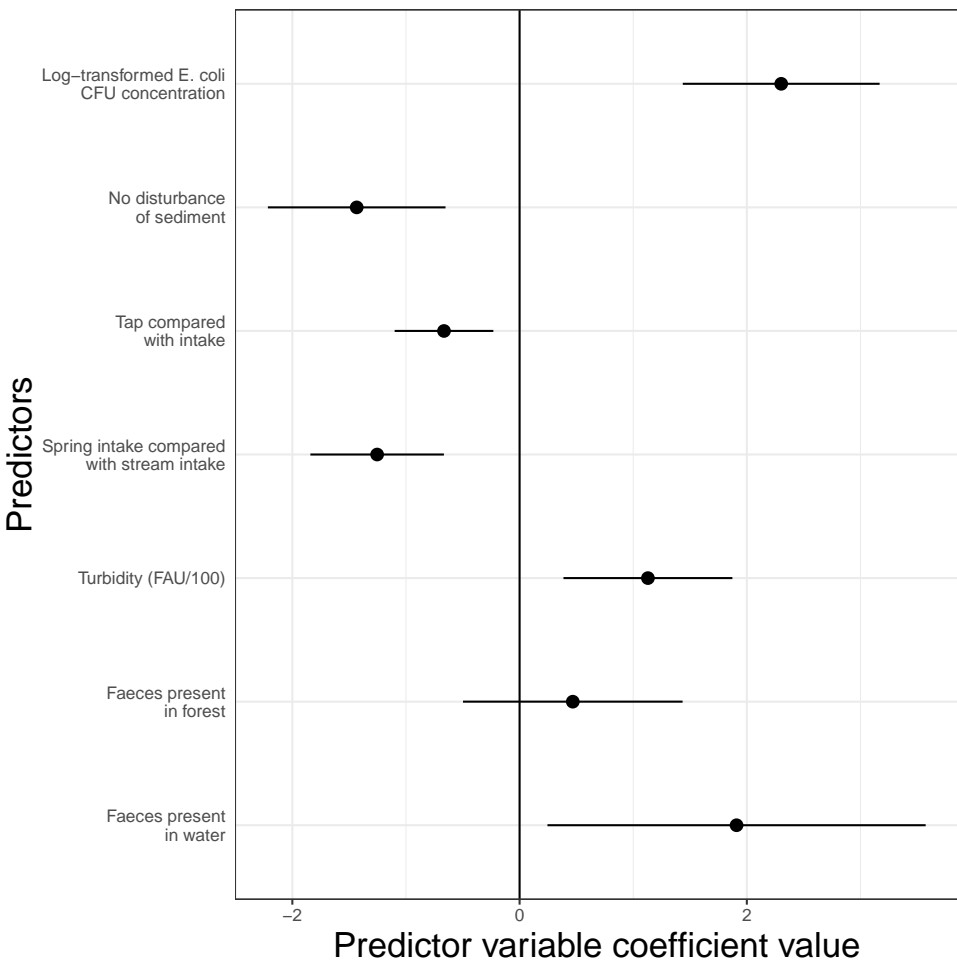

**Figure 4** **The effect of site features which predict 2015 *E. coli* concentration.** Error bars show 95% confidence intervals. This shows the results of the most highly supported GLMM (see Tables S4A–S4C for model selection tables and details of the presented model). Water systems $N = 124$.

sites with naturally higher contamination (e.g., streams rather than springs). Results from a negative binomial GLM show no significant effect of a site being in a control or treatment community on *E. coli* concentration in 2015. This is because there is no significant interaction between RCT status and *E. coli* concentration in 2010, meaning that the rate of change in *E. coli* concentration between 2010 and 2015 is not significantly different in sites associated with treatment or control communities.

In the second analysis we analyse *E. coli* at end-line only (levels in 2015) again using a negative binomial GLM (sites $N = 228$, communities $N = 116$). As we do not have baseline *E. coli* counts for all these sites we include important predictors of water quality from Fig. 4. The results show that while taps had significantly lower contamination than intakes and springs lower contamination than streams there was no significant effect of treatment (Fig. 6). Taken together both analyses together show that, using the robust RCT

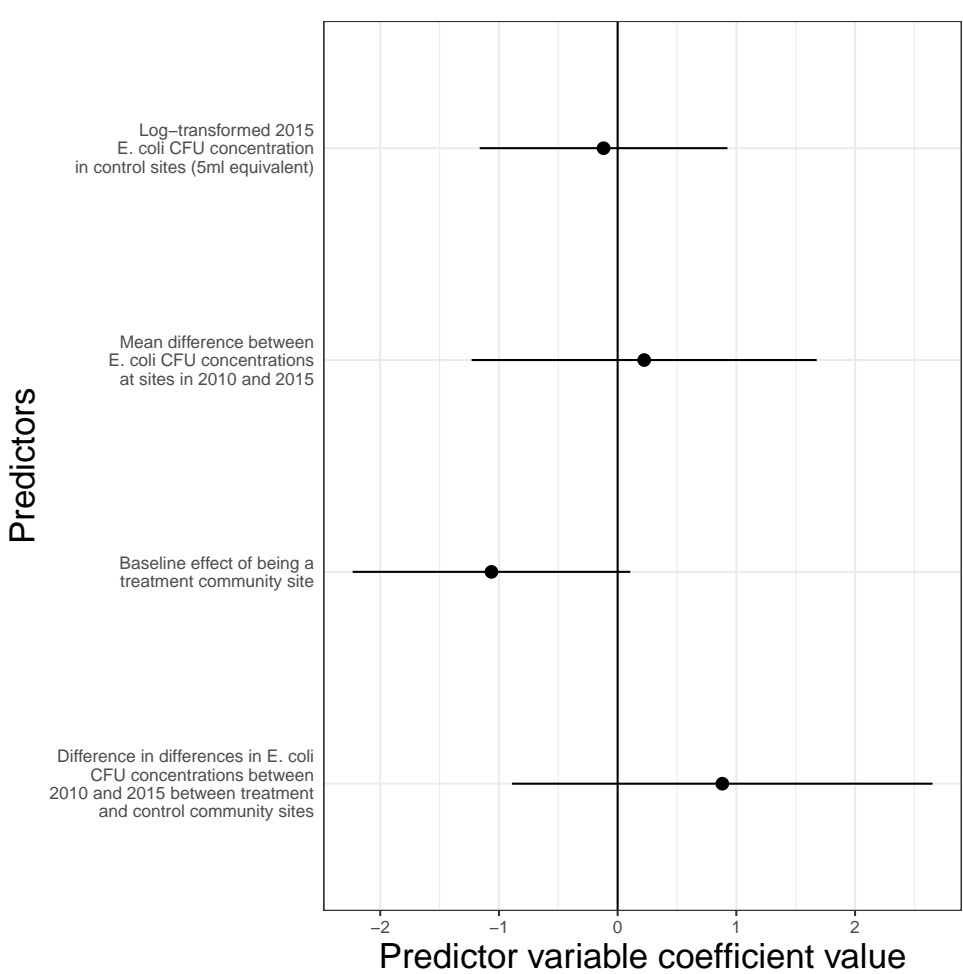

**Figure 5** **RCT difference-in-differences analysis matched on baseline *E. coli* contamination.** This analysis includes only the subset of communities where the sampling location remained the same between baseline and end-line (communities $N = 47$). The model shows no effect of the intervention on microbial water quality (the error bars on the coefficient 'difference in difference in *E. coli* CFU concentration between 2010 and 2015 between treatment and control sites' overlaps zero). The coefficients and confidence intervals are presented in Table S5.

design and both the baseline and end-line datasets, we did not find a significant effect of the intervention on *E. coli* concentration at the landscape scale.

## Uptake was low, highly variable and treatment and control communities do not differ with respect to protection of intakes from cattle

In treatment communities (where every household was offered the opportunity to enrol land) a low proportion of land eligible to be enrolled in *Watershared* agreements was actually enrolled, and this proportion was highly variable between communities (Fig. 7: range from 0–18% with a median uptake of 2.5%). There is no significant difference ($N = 129$; $p = 0.97$; chi-squared test) between the number of intakes protected from cattle
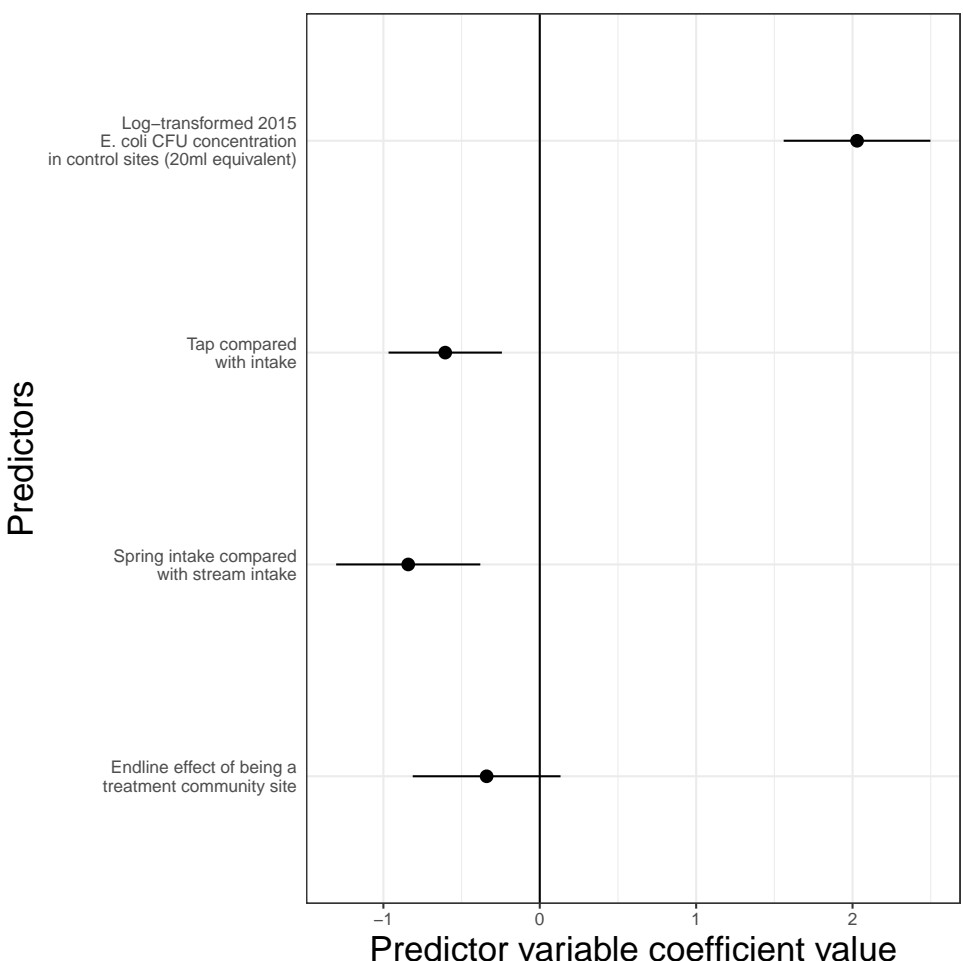

**Figure 6** RCT analysis for end-line data only from all communities for which we have end-line data (communities $N = 116$) controlling for predictors of *E. coli* contamination. This shows no effect of the intervention on microbial water quality (the error bars on the coefficient 'End-line effect of being a treatment community' overlaps zero). The coefficients and confidence intervals are presented in Table S6.

in control and treatment community sites. Water intakes in both control and treatment communities tend to be protected from cattle (61% and 62% of intakes respectively) despite only a quarter of intakes being in compliant level 1 areas (Table 2).

## DISCUSSION

### Did *Watershared* improve water quality?

We show that presence of cattle faeces in water or on the stream banks results in higher *E. coli* contamination at individual sites. This suggests that excluding cattle from water sources (one of the key actions *Watershared* seeks to incentivize) can contribute to improving water quality. This should perhaps not be surprising given that fresh cattle faeces can have more than $10^8$ kg$^1$ colony forming units (*Weaver, Entry & Graves, 2005*). However, the presence of cattle faeces is only one predictor of water quality. Intakes fed by streams were much

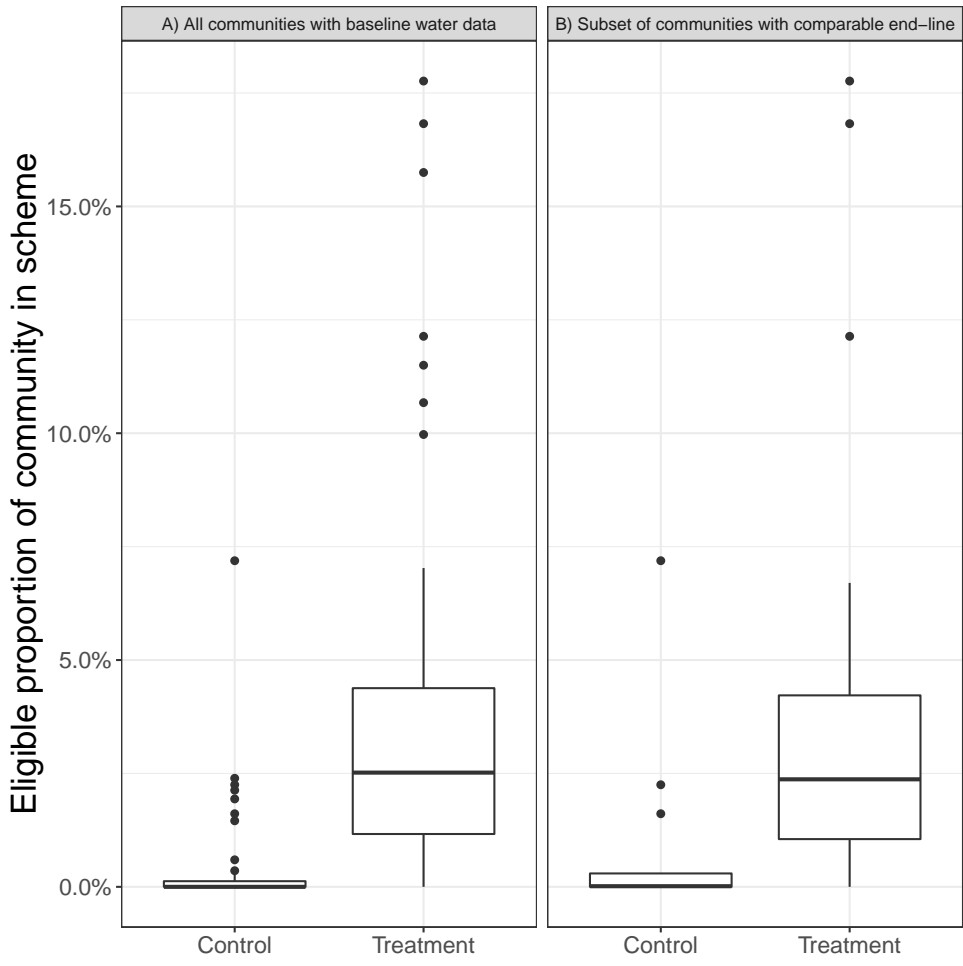

**Figure 7** **Box plots showing the difference between control and treatment communities in the proportion of eligible land enrolled in level 1 *Watershared* agreements.** The data is shown for all communities for which baseline water quality is available and the subset of communities for which we have a directly comparable end-line.

more contaminated than those fed by springs, as could have been predicted (*Howell, Coyne & Cornelius, 1995*; *WaterAid, 2013*). Indeed, the Millennium Development Goal definition of an improved water source allows some springs to be considered improved without further chemical treatment while no stream or river intakes can (*Bain et al., 2014b*). We also found that intakes were more heavily contaminated than taps. This suggests that although the sedimentation and filtration chambers in many of the water systems may not always be effective, they have at least some positive effect on water quality. It is unsurprising that turbidity was an important predictor as this is a well-known as a predictor of *E. coli* contamination (*LeChevallier, Evans & Seidler, 1981*).

Despite finding evidence that excluding cattle from riparian forests had a significant impact on water quality at the local scale, both the RCT analyses (comparing *E. coli*

**Table 2  Number (and proportion) of intakes visited in 2015 in compliant *Watershared* agreements and protected from cattle compared between treatment and control communities.**

|  | Treatment community intake | Control community intake |
|---|---|---|
| *N* | 68 | 61 |
| Compliant level 1 *Watershared* conservation agreement (%) | 16 (24%) | 0 (0%) |
| Sites with no cattle access (%) | 40 (59%) | 37 (61%) |

contamination between treatment and control communities) showed no evidence of an effect at the landscape scale.

There is therefore an apparent paradox in that the intervention incentivised (excluding cattle) does apparently improve water quality but the program has not had an impact at the landscape scale. It is possible that there has not been sufficient time between implementation of the intervention and subsequent evaluation for differences to become apparent. Some of the *Watershared* areas included in the analysis had been enrolled in the latter part of 2014, hence in some cases only a few months before the end-line monitoring was undertaken. It is well known that *E. coli* can persist in freshwater sediments for long periods of time (*Pachepsky & Shelton, 2011*; *Cho et al., 2016*). However, we argue that features of the way in which the PES program was implemented in the area meant that it was highly likely that no difference would be detected between control and treatment.

Firstly, a very low percentage of eligible land was enrolled in *Watershared* agreements (the median uptake of enrolled land was 2.5% of eligible land in treatment communities). In addition, GIS analysis has shown that much of the land which was enrolled could not influence water quality at the intake or tap because it was situated in a different catchment (*Pynegar, 2018*). Livestock-derived *E. coli* can enter water intakes through a number of routes including overland flow and movement of groundwater (*Oliver et al., 2010*), and not solely through direct deposition which is what the *Watershared* agreements try to prevent. The small areas conserved at or above the intakes may well have reduced faeces presence and so reduced *E. coli* concentration at these sites. However, upstream or uphill of these intakes contamination may have continued to enter water bodies through multiple routes. Instructively, evidence from a 26-year-old conservation area in the community of La Aguada, near to our study area, shows that despite the 20% of the catchment nearest to the intake being under conservation with cattle excluded, the water remains contaminated (*Pynegar, 2018*).

Secondly, the iteration of the *Watershared* intervention that we studied did not oblige, or even provide extra incentives, for landowners to conserve land surrounding or in the same catchment as monitored intakes. Farmers were free to enrol any land which met the criteria (forest within 100 m of a stream or spring). The intervention was not spatially targeted towards areas critical for community water supplies, and in fact only 16 of the 68 water intakes in treatment communities were located inside enrolled and compliant parcels of land. Also, many communities in the study area had previously excluded cattle from

water intakes independently of the *Watershared* program, meaning there is no significant difference in the proportion of intakes protected from cattle between treatment and control communities.

Finally, not all the land enrolled in *Watershared* agreements will represent additional conservation. There is a large body of evidence showing that adverse participant selection in PES programs means that a significant proportion of land enrolled would have been conserved in the absence of the incentives (*Börner et al., 2017*). There has been concern about the extent to which conservation funded under the *Watershared* programme will represent additional conservation since the early days of the scheme (*Robertson & Wunder, 2005*; *Asquith, Vargas & Wunder, 2008*). Best estimates for the *Watershared* intervention suggest that only about one third of level 1 agreements have resulted in cattle being excluded from land which they would have otherwise been using (*Bottazzi et al., 2018*).

Given the very low uptake of *Watershared* agreements, the lack of targeting of the land enrolled, and the fact that (like in any PES programme) not all the land enrolled represents additional conservation, it is perhaps not surprising that we did not detect an impact of the intervention on water quality at the landscape scale.

## How would *Watershared* have to change to result in a significant impact on water quality?

Some of the reasons why *Watershared* did not produce landscape-scale impacts on water quality relate to commonly recognised issues in PES implementation. First, the link between the land use incentivised (the proxy) and the ecosystem service desired is often weak and poorly understood (*Jack, Kousky & Sims, 2008*). In the case of *Watershared*, it is unclear how much land in a catchment would need to be protected, where, and over what timescale, to obtain a significant improvement in water quality at the landscape or even the catchment scale. Second, the marginal benefits from service provision (or in this case the land use proxy for service provision) are highly spatially heterogeneous. Land enrolled directly upstream of intakes will probably have an effect on monitored water quality while areas under conservation elsewhere (for example below the water intake) obviously cannot. In such cases, spatial targeting and differentiated payments would likely increase program efficiency (*Ezzine-de Blas et al., 2016*).

Both theory (*Persson & Alpízar, 2013*) and empirical research on PES programmes (*Arriagada et al., 2009*) suggest that low levels of payments result in low uptake. It thus seems likely that higher *Watershared* payments would have ensured that a higher proportion of eligible land was enrolled. It is difficult to directly compare payments in *Watershared* with those in related programmes both because of the payment structure (in *Watershared* participants are paid an enrolling fee plus a per hectare payment) and because a dollar is worth more in some countries than others. However the per-hectare payments in *Watershared* are certainly lower than other payments for watershed services-type programs in Latin America: for example Mexico's PSA-H program pays 27 USD/hectare/year for primary forest and 36 USD/ha/year for cloud forest (*Muñoz Piña et al., 2008*) while Costa Rica's national PES pays 45 to 163 USD/hectare/year (*Wunder, Engel & Pagiola, 2008*). The Ugandan PES program analysed in the RCT by *Jayachandran et al. (2017)* paid landowners

28 USD per year per hectare of forest. *Bottazzi et al. (2018)* show that those signing *Watershared* agreements have multiple motivations for doing so. To ensure that areas most valuable in terms of their potential ecosystem service provision were enrolled would require targeting. However such targeting would increase the complexity and transaction costs of the program (*Jack, Kousky & Sims, 2008*) and poses potential issues in terms of perceived fairness locally.

There is also likely to be a limit to the impact that livestock exclusion can achieve, and this will depend on the extent to which faecal contamination derives from other sources such as wildlife, inadequate sanitation infrastructure, spreading of manure on agricultural land, or from open defecation. Those involved in promoting similar interventions should check the extent to which cattle contamination is indeed the driver of microbial water quality issues in the region, perhaps using genetic testing of *E. coli* (*Carson et al., 2001*). Context-appropriate engineering solutions, such as protection of springs used for drinking water (*Kremer et al., 2011*), use of springs rather than streams as drinking water sources, construction of filtration systems, or introduction of household-level interventions (*Clasen et al., 2007*), may be more effective at improving water quality than livestock exclusions. Such solutions however do not provide the desired co-benefits of the intervention, such as forest carbon storage, biodiversity conservation, and increases in local incomes. Future work may aim to combine both conservation and engineering solutions and involve more direct conservation actions such as purchase or rent of particularly sensitive or important catchments.

### Did the RCT enable robust evaluation of the efficacy of the *Watershared* program?

Uptake of *Watershared* agreements was very low (the median enrolment of eligible land in treatment communities was just 2.5%). There was also no targeting of the land enrolled meaning that little of the land enrolled could impact water quality. Although 24% of the water intakes in treatment communities were in compliant level 1 *Watershared* agreements, there is no difference in the percentage of water intakes protected from cattle in treatment and control communities. This is because communities protect their water intakes for many reasons and it appears that the intervention was not significantly effective at increasing this level of protection. Therefore, the conclusion we draw from the RCT analysis about the efficacy of *Watershared* as delivered in this area at influencing water quality at the landscape scale, could equally have been drawn without a large-scale experiment conducted over 5 years. Theory-based impact evaluation (mapping out the causal chain from inputs to outcomes and testing the underlying assumptions; *White, 2009*) could have been just as effective. The RCT in this case therefore added little to evaluating the impact of this intervention.

Despite the challenges of implementing this RCT, and observation that the same conclusions about programme impact could probably have been made without the RCT, we remain positive about the potential value of RCTs in evaluating the impact of landscape-scale conservation. There is certainly demand from policy makers and funders for high quality causal inference about the efficacy of a programme and this is only going to increase.

We therefore suggest that those interested in quality impact evaluation of landscape-scale conservation interventions learn from the experience of the *Watershared* RCT.

## Lessons learnt from the experience of the *Watershared* RCT for future evaluations of landscape-scale conservation interventions

### The randomisation unit needs to be carefully selected

The choice of randomisation unit in an RCT will influence spillover; the phenomenon in which treatment may affect outcomes in non-treated units resulting in an erroneously low estimate of treatment effect size (*Glennerster & Takavarasha, 2013*). In the case of the *Watershared* RCT, randomisation was at the level of the community. However for evaluating the impact of the program on water quality, a more appropriate randomisation unit would have been the catchment above the intake supplying water to a community. The selection of communities as the randomisation unit in this RCT was problematic because catchments above treatment community water intakes could fall partially within neighbouring treatment communities (or vice versa); potentially resulting in spillover of effects between treatment and control communities (*Pynegar, 2018*). However *Watershared* was not designed only to impact water quality but also to reduce deforestation, improve forest biodiversity and provide socio-economic benefits (*Asquith, 2016*). Designing an RCT to evaluate the impact of an intervention on multiple outcomes is inevitably challenging (*Pynegar, 2018*). A future RCT of landscape-scale environmental management interventions should ensure that the randomisation unit is selected to minimise the risk of spillover. This may mean an RCT can only effectively evaluate the impact of one or two outcomes at once.

### There are difficult decisions to make in selecting sites at which to monitor outcomes

One of the aims of *Watershared* was to improve the quality of water consumed in communities. The decision was therefore made to measure water quality at the intake and tap serving the largest number of households in each community. Unfortunately, in a significant number of communities, the water intake serving the community had changed over the 5 years of the intervention. At such sites, if end-line data were to be taken from the water system serving the majority of the community, the end-line and baseline data were inevitably not at the same location. These sites thus had to be excluded from the difference-in-differences analysis. By conducting the analysis in two ways (our difference-in-differences analysis for the sub-set of communities where the intake was comparable across the period and using end-line data only from the full set of communities) we were able to use as much of the data as possible.

### An RCT is not appropriate unless the intervention is well developed

The substantial investment of time and resources in an RCT means that it is only appropriate when implementers are confident that they have an intervention whose functioning is reasonably well developed (*Pattanayak, 2009*; *Cartwright, 2010*). At earlier stages of developing an intervention, formative rather than summative impact evaluation processes might be more appropriate (*Rossi, Lipsey & Freeman, 2004*). The implementers of *Watershared* had experimented with versions of the program before implementing

the RCT. However in the event, uptake of *Watershared* was so low that an RCT was not an appropriate method of evaluating its impact on water quality at a landscape scale. Precise levels of uptake cannot be known until the experiment is already set up and the intervention offered, but pilot work (or analysis of the GIS data showing the location of enrolled agreements) might have revealed this low uptake and efforts been made to increase it (by raising the payments offered).

### Effort should be made to ensure balance between treatment and control in outcomes of interest

In the *Watershared* RCT, the allocation to treatment and control achieved quite good balance in the variables likely to influence *E. coli* contamination (although when only the subset of sites for which there is a comparable end-line is considered a significant imbalance in baseline values for *E. coli* contamination between treatment and control was introduced). Of course imperfect balance is a common problem in RCTs and is not a barrier to valid inference (*Senn, 2013*). We were able to account for the imbalance in our difference-in-differences analysis by matching on baseline *E. coli* values at each site. However to reduce the chance of random allocation to control and treatment resulting in an imbalance in important variables, baseline information on the outcome of interest (in this case *E. coli* contamination) should be included in the stratification.

### Care needs to be taken to avoid contamination of the control

In an ideal RCT, there is zero uptake of the intervention in the control group and consistent and high uptake in the treatment community. Unfortunately, in many situations the control is contaminated by information from treatment participants spreading to controls (*Torgerson, 2001*). In the *Watershared* RCT there was non-zero uptake among the control communities. In this case, the contamination of the control was due to people living in treatment communities but owning land in treatment communities which they chose to enrol. This was not noticed during the roll out of the program for two reasons. Firstly, the community boundaries were not available until they were generated (with substantial investment) by our research team in 2017. Secondly, for the technicians involved in day-to-day implementation of *Watershared*, the detail of where enrolled land occurred relative to treatment and control communities was not a priority. In future, we suggest that those establishing an RCT to evaluate the impact of environmental management interventions at scale ensure they have sufficient research capacity during the implementation phase to monitor the RCT while it is in progress so issues such as this could be avoided.

### Blinding is unlikely to be fully possible in land-scale scale conservation interventions

Double blinding is considered best practice in RCTs so neither the researcher nor the participants know who has been assigned to the treatment or control group (*Glennerster & Takavarasha, 2013*). In the case of the *Watershared* RCT the researchers scoring the *E. coli* contamination did not know whether the sample came from a treatment of control community; so blinding was achieved there. However the participants clearly knew they were in a control or treatment community. Those allocated to control or treatment may

have different expectations or show different behaviour or effort simply as a consequence of being allocated to a control or treatment group (*Chassang, Padró i Miquel & Snowberg, 2012*). Some authors have claimed that these behavioural effects may be large (*Bulte et al., 2014*) but they have not been extensively studied. They should be considered in the design of any landscape-scale RCT.

## CONCLUSIONS

There is global interest in PES because it is seen as an efficient way to provide environmental outcomes. The effectiveness of PES in achieving its intended outcomes is fundamentally an empirical question but the quality of the evidence base concerning the delivery of benefits from PES is mixed. There is therefore substantial interest in robust evaluation of the effectiveness of PES programs at delivering outcomes. We conclude that this particular program would require much greater uptake (probably requiring higher payments) and more intensive targeting (which would increase substantially the transaction costs and design complexity of the intervention) to have a significant impact on water quality. However, although this paper presents the results of a Randomised Control Trial (one of the very few implemented to evaluate the impacts of a conservation intervention at scale), these same conclusions could have been drawn without the RCT. The low uptake of the program and the lack of a difference in water intakes protected from cattle between control and treatment communities make the result of the RCT (no effect of the program detected at the landscape scale) inevitable. Randomised Control Trials have the potential to contribute to building the evidence base for understanding the impact of environmental management approaches such as Payments for Ecosystem Services. However, as evidenced by the *Watershared* experience, they are not straightforward to implement in practice. We hope that by publishing the experience of the *Watershared* RCT we will encourage future landscape-scale conservation impact evaluations to improve on the use of this evaluation approach in conservation.

## ACKNOWLEDGEMENTS

We thank colleagues at Fundación Natura Bolivia for all of their assistance, contributions and ideas, especially María Teresa Vargas, Tito Vidaurre, Maximo García, Dionicio Toledo, Hugo Vallejos, Miler Guzmán, Antonio Daza, Denis Calderón, Huascar Azurduy, Veronica Chávez and Victoria Aguilera. We are very grateful to Kelsey Jack for initial randomisation and to Dave Chadwick, Prysor Williams, Patrick Bottazzi, Robert Rueda and David Crespo for valued discussion. Sven Wunder and two anonymous reviewers gave very helpful feedback during the review process which substantially improved both the analysis and the framing of this paper. We also thank members of the communities in which we conducted monitoring for their support and permission.

### Funding

This work was supported by the Leverhulme Trust [grant RPG-2014-056], the UK's Ecosystem Services for Poverty Alleviation program [grants NE/I00436X/1 and NE/L001470/1], and a Doctoral Training Grant from the UK's Natural Environment Research Council [grant 1358260]. James Gibbons received financial support provided by the Welsh Government and Higher Education Funding Council for Wales through the Sêr Cymru National Research Network for Low Carbon, Energy and Environment. The funders had no role in study design, data collection and analysis, decision to publish, or preparation of the manuscript.

### Grant Disclosures

The following grant information was disclosed by the authors:
Leverhulme Trust: RPG-2014-056.
Poverty Alleviation program: NE/I00436X/1, NE/L001470/1.
Natural Environment Research Council: 1358260.
Welsh Government.
Higher Education Funding Council for Wales.

### Competing Interests

Dr. Nigel M. Asquith was the policy director of the NGO Fundación Natura Bolivia which implements the Watershared agreements evaluated by this paper. He was involved in designing the RCT which underpins the study, but did not collect or handle any of the data and was not involved in the analysis. Dr. Edwin L. Pynegar conducted this research as an independently funded PhD. student, but has since worked for Fundación Natura Bolivia as a consultant to help them design a future RCT to be undertaken in a different area of Bolivia.

### Author Contributions

- Edwin L. Pynegar conceived and designed the experiments, performed the experiments, analyzed the data, prepared figures and/or tables, authored or reviewed drafts of the paper, approved the final draft, mapping of sites.
- Julia P.G. Jones and James M. Gibbons conceived and designed the experiments, analyzed the data, prepared figures and/or tables, authored or reviewed drafts of the paper, approved the final draft.
- Nigel M. Asquith conceived and designed the experiments, contributed reagents/-materials/analysis tools, authored or reviewed drafts of the paper, approved the final draft.

### Data Availability

The raw data and data coding files are provided as Supplemental Files.

## Supplemental Information

Supplemental information for this article can be found online at http://dx.doi.org/10.7717/peerj.5753#supplemental-information.

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
