# Peer review of "The effectiveness of Payments for Ecosystem Services at delivering improvements in water quality: lessons for experiments at the landscape scale"

_PeerJ, doi:10.7717/peerj.5753_

## Round 0.1 · original submission · Major Revisions

[Following an Appeal of the decision by the authors and after discussion at PeerJ, we have revised this decision from Reject to Major Revisions, allowing the authors to revise the manuscript and address the issues raised. Revised editorial comments are provided below.]

The paper is well-written, organised and presented, and covers an important area of environmental research. It presents a novel use of a RCT to assess the effectiveness of PES for water quality improvement. However, unfortunately this statistical aspect has significant issues, which will need to be addressed before the paper can be considered for publication – since this pertains to research design as well as the analysis performed and its presentation.

Issues include:
- The use of the RCT, but with a very small sample-size and questionable data quality;
- The lack of comparability between treatment and control;
- The lack of precision in the RCT performed;
- The communities are not blind as to whether they or other communities are in the treatment or control group, leading to potential bias; while this is acknowledged, it compromises the statistical rigour of the research and at least requires exploration;
- It looks likely that communities are not independent – i.e. treatment in one community may affect the water intake quality in those nearby (although, it is difficult to tell with the way the sites have been presented);
- Supporting data/ information about communities is lacking.

I recommend that the focus of the paper be shifted from the use case towards a methodological assessment of the potential to use RCT in evaluating conservation interventions. The challenges and limitations of this approach should be addressed in some detail, particularly from a statistical standpoint, including issues with data and sample size, and the inconclusive or unreliable results which may be obtained from a small sample. If these issues are addressed, I believe that the paper could become acceptable.

· Appeal

Appeal


· · Academic Editor

Reject

The paper is well-written, organised and presented, and covers an important area of environmental research. It presents a novel use of a RCT to assess the effectiveness of PES for water quality improvement. However, unfortunately this statistical aspect has significant issues, which will need to be addressed before the paper can be considered for publication – since this pertains to research design as well as the analysis performed and its presentation, I have decided to reject the paper for publication at this time, but would encourage the authors to revise and resubmit.

Issues include:
- The use of the RCT, but with a very small sample-size and questionable data quality;
- The lack of comparability between treatment and control;
- The lack of precision in the RCT performed;
- The communities are not blind as to whether they or other communities are in the treatment or control group, leading to potential bias; while this is acknowledged, it compromises the statistical rigour of the research and at least requires exploration;
- It looks likely that communities are not independent – i.e. treatment in one community may affect the water intake quality in those nearby (although, it is difficult to tell with the way the sites have been presented);
- Supporting data/ information about communities is lacking.

The presentation of the manuscript is also unusual for this type of study – rewriting the paper more in line with environmental economics is recommended; alternatively, I recommend that the research design should be revised if it is not possible to overcome the limitations with the RCT conducted.

# ·

Basic reporting

Looks good to me; the text is well written and the arguments well articulated.

Experimental design

- Initially, E. Coli bacteria spread is strictly speaking a “brown” (pollution) problem, not related to the ecosystem functioning – i.e. arguably it is not an ‘environmental service’. Real env services (e.g. decompactation of soils, erosion control, etc.) might involve longer time lags. E. coli might this be seen as a bit of a canary in the coalmine for the impacts of this type of PES – or for the lack of them…

- E coli, treatment vs control: It may be a problem in the RCT evaluation method if the key output variable is skewed between sample and control -- that is randomization has not been effective in creating similar distributions.... The author cite a source that this is not a problem, but there are different opinions about that in the literature. There may be other systematic biases being introduced. It is quite weird to see in Figure 3 that for both 2010 and 2015 not even the whiskers overlap! Upfront, this raises the question how the random selection was in practice being done: some sort of lottery? Second, authors may need to do post-RCT matching of the sample before they compare the results (at least, also reporting this would increase confidence in the results).

- Treatment and control samples: were there no attrition communities, i.e. where communities resisted PES implementation? Zero attrition would be surprising. Please discuss.

- Similarly, the different levels of sample selection (and possible selection biases) are important to keep in mind: a) communities, b) landowners, c) land allocated by individual landowner. While for a) there is an RCT design, at the levels of b) and c), there is likely to be strong presence of adverse selection bias (ASB), i.e. ‘choosing the wrong parcels’ vis-à-vis additionality. ASB as a phenomenon is not much discussed here. For instance, who refused/ abandoned contracts within the selected communities (attrition)?

Validity of the findings

A lot of the interesting findings in the article are about PES effectiveness. I am integrating knowledge from reading Botazzi (2018) -- in order to better understand the scheme and the analysis of its results:

- upfront, it is remarkable that in a naive 'before-after' comparison, E Coli level go so much up, also in control communities.

- on that note, L.146-8 say: “Consent to randomization was granted by community leaders on the understanding that should the program be found to be effective, it would subsequently be implemented in all communities (this general roll-out was subsequently conducted in 2016).” But the results presented here can sow doubt about to what extent the approach was indeed effective.

- “A recent analysis demonstrated that the scheme has been 170 successful at incentivising additional conservation (Bottazzi et al., 2018).” Looking further into the evidence presented, the levels of additional conservation were not that high.

- Furthermore, self-reporting of respondents, done with Fundacion Natura paid interviewers, likely underreporting of e.g. non-additionality, if interviewers are perceived as non-fully independent of implementers.

- Apparent ‘micro-macro effectiveness/ additionality paradox’: the scheme is supposed to have been effective in terms of additional conservation, especially so for cattle retrieval (Botazzi et al. 2018). But at the same time, there is no effect at the landscape level (E Coli levels in communities).

- Possible explanations of ‘paradox’:
a) too little land area contracted (size)
b) too little focused on env sensitive areas (targeting)
c) mismatches between ‘action’/ proxy (cattle retrieval) and ‘results’ (E Coli content)
d) time lags between proxy and results
e) intra-community leakage effects from contracted to non-contracted lands

=> the draft so far focuses on a), b), c), d) (great discussion of those!), but maybe leakage also plays a role? Intra-community leakage: Did contracted landowners perhaps move cattle to other riverine areas – their own, or neighbours’ plots? Since only small land areas were typically enrolled, compared to the total of ‘suitable’ (target) land classes, this raises the risk of leakage…

However, the authors should also consider more about the option that ‘there is no paradox’, because the intervention is also not really effective at the micro level (challenging the conclusion in Botazzi et al. 2018), due to:
f) lack of careful spatial targeting
g) self-declared additionality levels of the ‘action’ is too low to make a community-level difference;
h) those self-declared levels are still upward biased by respondents’ drive to please (F. Natura employed) interviewers re. project impacts

Finally, while I have deep respect for Fundacion Natura's efforts and experience in conducting and replicating PES schemes, and think the RCT design is a great idea, the question arises: How to run a 65 community PES schemes as a single NGO without losing your sanity? Implementing a PES scheme requires significant effort: how can one do that seriously in 65 communities at the same time? E.g. what about quality of the design (spatial targeting etc.) and compliance (monitoring and sanctioning) system, outlined recently as key for PES functioning (Wunder et al 2018)? Where so many corners necessarily cut, so that PES effectiveness would have to suffer?

Reviewer 2 ·

Basic reporting

The manuscript is polished and reads clearly. Its excellent exposition is one of the greatest strengths of the paper. The motivation for the paper is clear, the approach is unique, and the paper is certainly likely to make an impact in the field.

Perhaps my only concern is that I seem to be missing a table with results? Or perhaps those results are only in the supporting information? The authors might want to consider including at least one table of results in the main text, although now that I think about it, perhaps the authors did this intentionally to illustrate their findings with figures rather than tables?

Experimental design

The authors do a very good job describing what was done, and what the limitations were with the study.

Water quality monitoring for 241 sites is a massive undertaking. I am familiar with the Coliscan Easygel method and agree that it is appropriate for this context. The authors recognize different procedures in baseline and endline data collection, but justify this and explain it well.

Validity of the findings

The authors eliminate sites from their sample if they were unable to verify T & C consistency in the two sampling periods. But could there have been systematic bias that eliminated sample sites? Were there issues of non-compliance?

What about seasonal variation in which samples were taken?

The results are not what one would like to see as a researcher, but the authors do a good job of pursuing the topic a bit further with the second set of models. This only produces a somewhat unsurprising result, but in a sense, it confirms the justification for the program in the first place, so the two analysis work well with one another.

Additional comments

The paper reads very well, and is an innovative piece of research. It certainly will be well received.

It seems odd that there are no Bolivians included as authors? Especially given the length and intensive fieldwork of this study. It seems odd, and does not reflect well in my opinion.

Lastly, one minor editing suggestion, consider rewording the sentence on line 466. Otherwise, it is a brilliantly written piece.

Reviewer 3 ·

Basic reporting

Although the research question is very interesting, I find the paper poorly motivated. The reasons why we need to evaluate the effectiveness of PES (in general and here in particular) are not given in the introduction, and yet there exists a variety of reasons for a PES not to be effective in changing practices. A discussion about self-selection in voluntary programs and potential spillovers is needed.
The motivation of the program under study is also unclear: why do we have to pay people to stop polluting the river with their livestock? They are the first victims of this pollution; they should not need to be paid for it. More background about the program is needed.
The literature mentioned is relevant but the main findings of these studies are not given; it is thus difficult to evaluate the contribution of the paper.
Overall, the paper does not look like a standard “RCT paper”, like those that can be found here for example: https://www.povertyactionlab.org/evaluations Many information is missing. I would suggest completing the manuscript following the standard approach used in papers in economics.

Experimental design

The study relies on a randomised controlled trial, which is gold standard in the evaluation of treatment effectiveness when well designed and implemented. In this study however, the sample size is very small (n=123) and the control and treatment groups are not comparable ex ante, since E. coli concentration is higher in the control group. I does not mean that the data cannot be used (since the allocation of clusters between groups is carried out randomly, it is known that any differences that do occur must have occurred by chance (Hayes and Moulton, 2009)). However, the authors should at least provide balance t-tests of baseline observables in both groups, so that one can judge the seriousness of the problem. In particular, I would like to know the average number of cows per village, the average number of households that leave their cows in riparian zones, the average number of children with diarrhea, other sources of income (other than cattle ranching), etc.
The methodology used is called Difference-in-Difference but this is never mentioned in the paper.
The econometric part of the analysis that aims to correlate water quality and cattle presence is very unclear (for example, the econometric model used is missing) and does not bring much value to the paper I think, since it has already been established in the literature that the major source of water contamination is the presence of cattle.

Validity of the findings

The main result lacks precision, so that the authors cannot conclude about the effectiveness of the program. The sample size being very small, it would have been interested to calculate the minimum detectable effect (MDE).
The main outcome (water quality) is not measured in the same way during both surveys (monitoring is different in 2015; the location of taps is also different in 2015). The authors should explain to what extent these changes threaten the validity of the results.

Additional comments

The research question is potentially interesting but poorly motivated, so it is difficult to appreciate the contribution of the paper. In general, the paper is not written as a paper in economics, whereas it should, it seems to me, since it deals with changing the practices of breeders in Bolivia.
The statistical method used is good but the quality of the data (especially the main outcome) is poor and the sample is too small to show the impact of the program – if there is any. In the absence of significant results, it is difficult to see the interest of the paper.

---

## Round 0.2 · Minor Revisions

Thank you for taking the time to revise your manuscript. Your rewrite and reframing of the presentation has transformed the paper: it now reads as an open and honest appraisal of the challenges of RCT in the evaluation of PES. I will be happy to recommend publication of your article since I believe that it will be of help to other researchers in efforts to implement RCT or other forms of evaluation. However, first there are a few minor suggestions which I think can further improve the paper:
1) As suggested by Reviewer 3, please consider including the Heard et al. 2017 paper as part of the discussion. I suggest you evaluate the challenges and solutions presented by Heard et al. within the context of your research programme and PES more generally. A summary table may assist with this.
2) Note the comment by Reviewer 3 regarding minimum detectable effect: please comment on this and include a sentence in the paper.
3) Check the minor comments from Reviewers 1 and 2 and consider any edits necessary.
I expect that these changes will be straightforward and quick to implement. I look forward to receiving the revised manuscript.

·

Basic reporting

Fine!

Experimental design

A much better description than in the first submission.

Validity of the findings

I think the authors have done a very good job in using the reviewer comments to significantly improve the paper! This is now a very thoughtful and interesting contribution to the literature. I have no doubt in recommending it for publication in its current stage.

Some very minor issues that the authors may consider:

- It might be worth mentioning that F Natura's pilot scheme in Los Negros was also analyzed from a PES angle (referring e.g. to Asquith et al, 2008, EcolEcon), Already at that point the likely lack of additionality was alleged, due to low payments and lack of targeting (see also Robertson & Wunder, Fresh Tracks in the Forest, 2005 on this aspect).

- l.352, downweighted (join words)
- l.450-63, point on payment levels very well-taken -- just with the caveat that Mexico and Costa Rica (possibly also Uganda) are higher-income countries than Bolivia, where 1 USD has more purchasing power.

Additional comments

None

Reviewer 2 ·

Basic reporting

Once again, the authors provide a well written piece.

Experimental design

In my first review, I had fewer objections than others. The new version should address the concerns raised, as the responses are detailed. Some of the issue is just language, like the choice to use "difference in difference" - ecologists refer to this as BACI, etc...

I am not enthusiastic about the addition of a Matching approach (Ho et al). I understand why this is added, and it makes sense for those with sample sites, it us back to an "observational world", while the whole point of the RCT is to avoid going there.

Validity of the findings

The section on the lessons is very valuable, especially the observations pertaining to the randomization unit and using well developed interventions in RCTs. Overall, this section is very nice.

Additional comments

I recommend publication. This paper must have been a monumental undertaking. It was clear that one of the other reviewers had no idea how difficult and enormous this water monitoring project must have been. My only regret is that credit could not have been shared with Bolivian co-authors, but of course the authors are best positioned to make such decisions.

It is a fine piece of research.

Reviewer 3 ·

Basic reporting

no comment

Experimental design

no comment

Validity of the findings

In this new version of the paper, the authors acknowledge the main failures of their RCT: attrition, spillover, data quality, unit of randomization, etc. This makes the study more honest about the scope of the results.

In my previous report I asked the authors to provide the minimum detectable effect, which is the difference between the 2 groups that could have been estimated given the sample size (the smaller the effect we seek to recover, the larger the sample must be to have a chance to detect the effect); I did not find it the revised manuscript.

Additional comments

The authors have made the effort to be explicit about the limits of their design, which is a good thing in itself. On the other hand, it is now clear that the absence of conclusive results can be explained by the many weaknesses of the design of their RCT. I wonder then what is the real contribution of the article to the literature on the effectiveness of the PES. The authors have rewritten the paper by guiding the discussion on the lessons to be drawn from their failure, but this type of discussion has already been done, and in more depth in several papers, like the one by Heard et al. (2017) for example. I thus remain unenthusiastic about the publication of this paper.
Cited reference:
Kenya Heard, Elisabeth O’Toole, Rohit Naimpally, Lindsey Bressler (2017) "Real-World Challenges to Randomization and Their Solutions", J-PAL North America, April 2017

---

## Round 0.3 · accepted · Accept

This is a very nice paper - thanks for taking the time to reframe and substantially rewrite the manuscript. I am happy to see this published now: it raises significant questions about how we can evaluate effectiveness of PES, and I expect will serve as a valuable reference point for researchers, NGOs and other agencies.

#